# LAMP: LARGE MODEL PRUNING WITH INTER-BLOCK ERROR COMPENSATION

## ABSTRACT

The increasing prevalence of large-scale models, both in vision and language domains, presents significant challenges in terms of memory and resource consumption. While model pruning is an effective method for compressing models to alleviate these constraints, existing techniques either require extensive fine-tuning, which is resource-intensive, or perform well only at low sparsity levels $(10\% - 50\%)$, failing at high sparsity levels $(50\% - 90\%)$. To address these issues, this paper introduces LAMP to mitigate the drawbacks associated with traditional pruning methods, namely high resource consumption in methods that require extensive fine-tuning, and poor performance at high sparsity levels in methods that do not. It reduces memory overhead and alleviates performance degradation at high sparsity. Experimental results demonstrate that LAMP achieves slightly better performance than SparseGPT at low sparsity levels and significantly better at high sparsity levels in both language and vision models, without significantly increasing memory consumption when compared to SparseGPT.

## 1 INTRODUCTION

Since the seminal works by (Vaswani et al., 2017) on the Transformer architecture and (Dosovitskiy et al., 2020) on the Vision Transformer, there has been substantial progress in both vision and language models leveraging the foundational Transformer structure. In language modeling, model series such as OPT (Zhang et al., 2022) and Llama (Touvron et al., 2023a;b) have harnessed the Transformer architecture to construct large language models that deliver outstanding performance. Similarly, significant vision models like Segment anything model (SAM) (Kirillov et al., 2023) have employed the Vision Transformer to develop robust image representations, demonstrating excellent performances in tasks such as segmentation, classification, and various downstream applications (Ke et al., 2024; Wu et al., 2023).

These models are remarkably large in scale. For instance, the largest OPT model contains 175 billion parameters, while the latest Llama v3 (Llama Team, 2024) model boasts up to 405 billion parameters. The largest SAM model has 636 million parameters, which, although significantly smaller, still poses deployment challenges in certain scenarios. To facilitate the deployment of these large-scale models, model compression techniques are often employed to reduce their size, enabling them to meet hardware constraints.

Pruning is a critical model compression technique, often categorized into multi-shot and one-shot methods based on pruning frequency. Traditional iterative pruning methods, based on the Lottery Ticket Hypothesis (Frankle & Carbin, 2018), involve an iterative prune-and-finetune process to gradually identify a sparse subnetwork that performs comparably to the original model, thereby reducing the number of model parameters. Classic methods like magnitude pruning (Han et al., 2015b) and more recent approaches such as LLM-Pruner (Ma et al., 2023), which utilizes block pruning with LoRA (Hu et al., 2021) fine-tuning, adhere to this hypothesis. However, for large models, global fine-tuning is resource-intensive, demanding significant GPU memory. Although LLM-Pruner reduces memory requirements using LoRA, it still requires storage for intermediate results and gradients, while methods like LLM Surgeon (van der Ouderaa et al., 2023), despite avoiding global fine-tuning, also demand substantial memory and time due to the use of second-order derivatives. In contrast, one-shot pruning methods have gained traction by minimizing memory usage during fine-tuning. SparseGPT (Frantar & Alistarh, 2023) avoids global fine-tuning by compensating for

precision within each basic layer, thus accelerating the pruning process. Wanda (Sun et al., 2024) further simplifies this by eliminating compensation altogether, using a unique significance score to achieve performance comparable to SparseGPT without the need for fine-tuning.

However, both SparseGPT and Wanda have notable limitations. (i) While these methods perform well at lower sparsity levels (10%-50%) when pruning large language models, their effectiveness significantly declines at higher sparsity levels (50%-90%). (ii) SparseGPT and Wanda show poor performance on vision models. Wanda, in particular, relies on priors that are specific to language models, making it even less effective than SparseGPT for vision tasks.

Regarding these issues, in this paper, we propose LAMP, a large model pruning method that is applicable to both vision and language models. It improves performance at both low and high sparsity levels in pruning large language and vision models, without increasing GPU memory usage and pruning time. Specifically, we design a sliding window-based inter-block error compensation method based on the principle of error propagation. As illustrated in Fig. 1, after pruning each block, the subsequent block compensates for the pruning-induced errors of the previous block before pruning itself. This approach ensures that errors introduced during pruning are compensated for in the next block, preventing error propagation to deeper layers and avoiding the need for full-model fine-tuning during the pruning process. We highlight our contributions as follows:

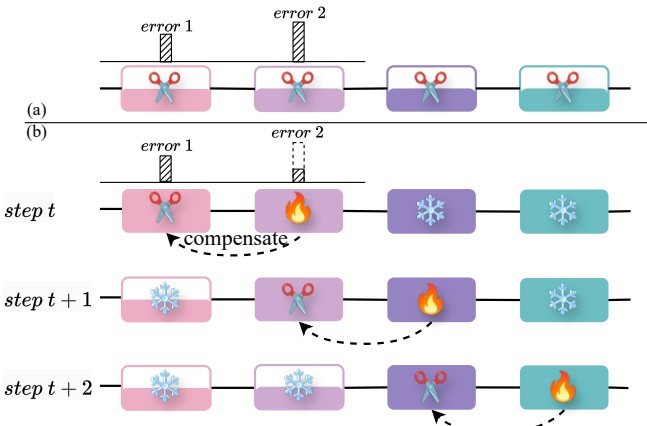

Figure 1: Comparison between our proposed method LAMP and conventional pruning methods. (a) illustrates a standard pruning approach, where all blocks are pruned simultaneously. (b) demonstrates our method. After pruning one block, the subsequent block adjusts to compensate for the error introduced by the pruning of the previous block.

1. We introduce the concept of inter-block error compensation, demonstrating that this approach can enhance the performance of one-shot pruning methods with limited computational resources.

2. Building on this concept, we further design a sliding window-based inter-block error compensation strategy. After pruning each layer, we use the next unpruned layer to compensate for the errors induced by pruning the current layer. It improves post-pruning model performance in both language and vision models.

3. Extensive experiments on the OPT series of large language models and the SAM series of vision models show that our method significantly enhances post-pruning model performance without increasing the GPU memory consumption.

## 2 RELATED WORK

### 2.1 TRANSFORMER ARCHITECTURE

Since its introduction (Vaswani et al., 2017), Transformer architecture has become the cornerstone of language models due to its superior performance in language tasks. Following the development of GPT (Radford et al., 2018), researchers have built increasingly large models using Transformer as primary building blocks. The Transformer has also proven effective in vision tasks. For example, SAM (Kirillov et al., 2023), which is based on the Vision Transformer, particularly excels in segmentation and serves as a foundational model for various downstream tasks (Wu et al., 2023; Mazurowski et al., 2023; Quan et al., 2024).

Both language and vision models utilize a backbone composed of multiple Transformer blocks. The majority of parameters in these models are concentrated within their respective backbones. These Transformer blocks consist of multi-head self-attention and feed-forward neural networks with two linear layers. Multi-head self-attention is generally considered more critical than the feed-forward network.

## 2.2 LARGE MODEL PRUNING

Prior to the advent of large models, the development of pruning methods primarily aimed to enhance model performance at target sparsity levels, leading to many highly effective techniques. Pruning methods can be categorized by granularity into structured and unstructured pruning. Specifically, unstructured pruning (Han et al., 2015b) modifies individual weights, generally achieving better performance but requiring a sparse engine for inference acceleration on GPUs. In contrast, structured pruning (Liu et al., 2017) operates on regular blocks of weights, allowing direct inference acceleration on GPUs, although it does not match the performance of unstructured pruning. Currently, popular pruning methods are typically iterative (Frankle & Carbin, 2018). These methods iteratively prune a portion of the weights, followed by global fine-tuning to restore performance, repeating this cycle until the target sparsity is achieved. While iterative pruning improves the performance of pruned models, it requires multiple rounds of fine-tuning, resulting in longer pruning times. Some research has also focused on one-shot pruning methods (Han et al., 2015a; Lee et al., 2018). These methods usually offer faster pruning and some do not require fine-tuning the entire model, thus saving significant memory costs. However, one-shot pruning methods typically do not achieve the same performance as iterative pruning methods.

With the emergence of large models, the computational and memory costs of global fine-tuning made iterative pruning methods prohibitively expensive. However, some approaches have combined iterative pruning with LoRA to apply this technique to large models (Ma et al., 2023; Zhang et al., 2023). Additionally, the speed and lower computational resource requirements of one-shot pruning methods have led to the development of several such techniques for large models, notably SparseGPT and Wanda. SparseGPT (Frantar & Alistarh, 2023) is a one-shot rapid pruning method belonging to Optimal Brain Surgeon (Hassibi & Stork, 1992) family. It independently analyzes each basic layer of the model, formulates the pruning problem as a constrained optimization problem, and solves the pruned weight values using the Lagrange multiplier method. SparseGPT builds on the OBS foundation by leveraging the property of Cholesky decomposition to efficiently obtain different orders of Hessian matrices, significantly accelerating the pruning process and reducing memory usage. Essentially, SparseGPT consist of two parts: pruning and compensation, which are integrated during the Lagrange multiplier method's solution process. However, SparseGPT compensates for pruning errors by adjusting unpruned weights within each basic layer, without considering inter-block weight interactions. Wanda (Sun et al., 2024), another pruning method, is faster than SparseGPT. It introduces a strong prior for language models, assuming that weight importance is related to both the absolute value of the weight and the corresponding output. Wanda designs an importance metric that achieves good results without compensating for pruning losses. However, subsequent studies Williams & Aletras (2023) suggest that while Wanda performs well in many cases, it can occasionally fall short compared to SparseGPT, and its performance in vision models is particularly lacking, where its strong priors may lead to suboptimal results.

## 3 METHOD

Our primary objective is to develop an efficient pruning method for large models, including both vision and language models. This method should be executable within a reasonable time frame, even under limited computational and memory resources. Additionally, it should sustain robust performance at high sparsity levels. To achieve this, we have carefully balanced the need for high performance with the imperative to minimize memory and time overhead. In this paper, we introduce LAMP, a pruning strategy that effectively mitigates performance degradation at high sparsity levels and facilitates rapid pruning in memory-constrained environments. The following sections detail our method through four key aspects: Theoretical Foundations, Progressive Block Pruning, Sparsity Rearrangement, and Chunked Intermediate Storage.

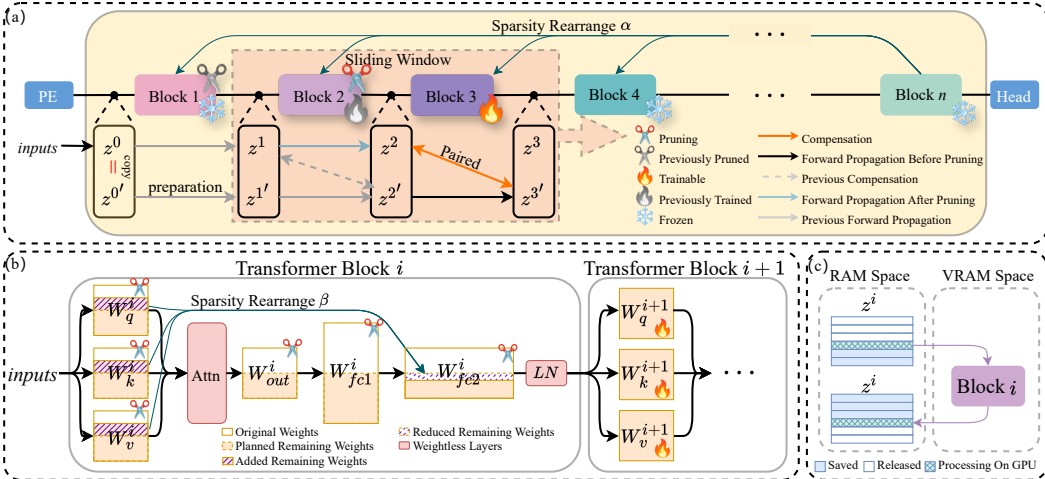

Figure 2: Overview of the proposed method. (a) progressive block pruning for sequential block pruning and compensation; (b) basic structure of a transformer block (showing only weight-intensive and essential layers); (c) illustration of the chunked intermediate storage.

## 3.1 PROGRESSIVE BLOCK PRUNING

As discussed in the introduction, traditional pruning methods typically require global fine-tuning to compensate for pruning-induced errors, necessitating the entire model to be loaded into memory. If full-precision fine-tuning is used, additional memory is needed to store full-precision gradients, resulting in a memory demand of at least twice the model's size, often more. While methods like SparseGPT avoid full-model fine-tuning, significantly reducing memory usage, they only compensate for errors within individual blocks and overlook inter-block interactions. Given that blocks function as nested entities, where earlier blocks influence subsequent ones, ignoring these interactions can lead to error accumulation. This issue may be negligible at low sparsity but becomes critical at high sparsity, leading to rapid performance degradation, as confirmed by our experiments.

To address this, we propose Progressive Block Pruning, which enhances inter-block error compensation by building upon SparseGPT. As depicted in Fig. 2(a), Progressive Block Pruning leverages the modular structure of large models, processing each block sequentially using a sliding window approach. Initially, the sliding window contains two blocks. Since the operations are consistent after each slide, we explain the method using the first two blocks.

In Fig. 2(a), the sliding window moves from left to right with a window length encompassing two blocks. Initially, the window contains $B^1$ and $B^2$. Pruning is first applied to $B^1$, followed by compensating for the pruning-induced error in $B^1$ using $B^2$. The detailed calculation procedure for this compensation is outlined in Algorithm 1, as follows: The input data from the calibration set passes through the position encoding module, producing an intermediate result, $z^0$, which is then duplicated to create $z^{0'}$. At this stage, the feature flow splits into two streams: the "standard stream" and the "error stream." Before pruning $B^1$, $z^{0'}$ passes through $B^1$, producing an unbiased output, $z^{1'}$. SparseGPT is then applied to prune $B^1$, and $z^0$ passes through the pruned $B^1$, resulting in a biased output, $z^2$. Subsequently, $z^{1'}$ passes through the unpruned $B^1$, yielding an unbiased output, $z^{2'}$. Here, $z^1$ serves as both the output of the pruned $B^1$ and the input to $B^2$. Since $z^1$ is biased, the output from $B^1$ will also be biased, whereas the desired output is $z^{2'}$. Therefore, $z^{2'}$ serves as the target for $z^1$, forming a labeled feature dataset $(z^1, z^{2'})$. To align output of $B^1$ with $z^{2'}$, we adjust $B^1$'s parameters by training it on this feature dataset, reducing the output bias. Finally, the sliding window advances by one block, and the process is repeated: prune the first block and adjust the parameters of the second block using the newly formed feature dataset. A brief derivation of the proposed method is provided in Appendix A for further details.

As above, Progressive Block Pruning progresses sequentially along the model's backbone, processing one block at a time. It only requires a single block to be loaded into memory at a given moment.

As each block is loaded into memory and pruned, the previous block is unloaded before the next block is loaded. This approach significantly reduces memory usage during the pruning and compensation process.

## 3.2 SPARSITY REARRANGEMENT

In many pruning methods, the sparsity of different parts of the model can be adaptively determined through specific rules. However, since our method is based on a sliding window approach for progressive pruning, it is difficult to obtain information about the blocks that are outside the current window. If we attempt to acquire global information, we would either incur significant I/O overhead from loading model segments sequentially or face high memory usage from loading the entire model at once. Some methods even evaluate importance based on gradients, which can add further demands on gradient storage.

To avoid these issues, fast pruning methods like SparseGPT or Wanda typically assign the same sparsity level uniformly across all parts of the model, equivalent to the target sparsity. However, this approach is clearly suboptimal. Certain prior knowledge can be leveraged to adjust the sparsity distribution, thereby avoiding these problems while improving performance.

---

**Algorithm 1** Progressive Block Pruning

**Input**: Backbone of large model $\{B^i\}_{i=1}^n$, i-th block $B^i$, Pruning function $Prune(module, hyperparameters)$, Calibration dataset $x_j$, Position embedding block $PE$, Update function $Update(module, (sample, label))$, Sparsity $s$, Sparsity Rearrangement parameter $\alpha, \beta$
**Output**: Pruned backbone $\{B^i\}_{i=1}^n$

1: $z^0 = PE(\{x_j\})$
2: $z_j^{0'} = z_j^0$ // Copy embeddings
3: $z_j^{1'} = B^0\left(z_j^{0'}\right)$ // Preparation
4: **for** $i$ in 1 to $n-1$ **do**
5:     $B^i = Prune(B^i, (s, \alpha, \beta))$
6:     $z_j^i = B^i\left(z_j^{i-1}\right)$ // Forward Propagation
7:     $z_j^{i+1'} = B^{i+1}\left(z_j^{i'}\right)$
8:     $B^{i+1} = Update\left(B^{i+1}, z_j^1, z_j^{2'}\right)$
9: **end for**
10: $B^n = Prune(B^n, (s, \alpha, \beta))$
11: **return** Pruned backbone $\{B^i\}_{i=1}^n$

---

### 3.2.1 INTER-BLOCK SPARSITY REARRANGEMENT

Progressive Block Pruning fine-tunes $B^{i+1}$ to compensate for the pruning-induced errors in $B^i$. However, when pruning the last block, $B^n$, there is no $B^{n+1}$ available to compensate for the errors in $B^n$. Therefore, assigning the same number of pruned weights to each block is not optimal. Instead, the pruning distribution should be adjusted across different blocks. To maintain the global target sparsity, we rearrange the sparsity between blocks as follows:

$$P_{B^n} = |\theta^n| \times s \times \alpha, \tag{1}$$

$$P_{B^j} = |\theta^j| \times s + \frac{|\theta^n| \times s \times (1-\alpha)}{n-1} \tag{2}$$

where $j = 1, \ldots, n-1$, $\theta^j$ denotes the parameters in $B^j$, $|\theta|$ denotes the number of weights in a single block, and $\alpha$ is a super-parameter that redistributes a portion of the pruned weights from the last block evenly across all preceding blocks.

### 3.2.2 INTRA-BLOCK SPARSITY REARRANGEMENT

As shown in Fig. 2(b), the majority of the parameters within a Transformer block are contributed by the matrices depicted, which are clearly hierarchical. Therefore, a block can essentially be viewed as a nested function: the QKV matrices represent the innermost function, while the second fully connected layer (fc2) represents the outermost function. According to the conclusions drawn in the Theoretical Foundations section, if the same sparsity level is applied uniformly across all layers, the errors introduced by the QKV matrices, which are further from the output of $B^i$, will be more challenging to compensate for in $B^{i+1}$. On the other hand, fc2 is closer to the output of $B^i$, making its errors easier to compensate for in $B^{i+1}$. Additionally, inspired by previous works, earlier layers tend to converge first (Chen et al., 2023), and faster convergence often leads to better generalization performance (Hardt et al., 2016). Therefore, the sparsity distribution should be rearranged by reducing the sparsity of the QKV matrices and increasing the sparsity of fc2. To ensure that the global target sparsity remains unchanged, we rearrange the sparsity according to the following rule:

$$P_t = s \times |W_t^i| \times (1-\beta), \quad t \in \{Q, K, V\}. \tag{3}$$

$$P_{fc2} = s \times |W_{fc2}^i| + s \times |W_{fc2}^i| \times \beta. \tag{4}$$

where $P_t$ represents the number of pruned weights, $|W|$ denotes the total number of weights in a single layer, and $\beta$ is a super-parameter that needs to be adjusted.

### 3.3 CHUNKED INTERMEDIATE STORAGE

Proposed method requires the intermediate results of each sample after passing through each block. Since the storage space required for these intermediate results is significantly larger than the space occupied by the samples themselves, simply storing them in GPU memory, as done in typical training processes, would consume a considerable amount of memory. Given that GPU memory is a more valuable resource than system memory, this approach would lead to significant memory wastage. To address this issue, we employ Chunked Intermediate Storage to reduce memory usage during the pruning process.

As in Fig. 2(c), all intermediate results are stored in system memory, and they are only loaded into GPU memory when needed for processing through a block. For instance, While the GPU processes $B^i$ and $z_j^i$, the CPU simultaneously loads $z_{j+1}^i$ into GPU memory and releasing it from system memory. After $z_j^i$ is processed and get $z_j^{i+1}$, it will be transferred from GPU memory back to system memory. This way, only two intermediate results and one block are kept in GPU memory at any given time, minimizing memory usage without affecting the speed of forward pass.

## 4 EXPERIMENTS

### 4.1 EXPERIMENTAL SETUP

To evaluate the effectiveness of our method, we focus on models with 7 billion parameters or fewer, as these are the most commonly used in research and resource-constrained scenarios. Pruning these models is also the most feasible on a single consumer-grade GPU. Specifically, we validate our approach using the OPT-125M/1.3B/2.7B/6.7B and Llama 2-7B language models from Hugging Face, as well as the SAM-B/L/H vision models provided by Meta. All experiments were conducted on a single NVIDIA RTX 3090 GPU with 24GB of memory.

For the hyperparameter settings of LAMP, learning rate is adjusted according to the sparsity level, as different sparsity levels introduce varying amounts of error. $\alpha$ and $\beta$ are tuned based on the training dynamics.

For calibration data, the language model calibration dataset follows the approach of SparseGPT, using 128 segments of 2048 tokens, randomly sampled from C4 dataset (Raffel et al., 2020). Similarly, the vision model calibration dataset consists of 128 images, randomly sampled from the second compressed file of the SA-1B dataset (Kirillov et al., 2023) provided by Meta.

To comprehensively evaluate the performance of pruned models, we validate OPT using the Wikitext-v2 (Merity et al., 2016) and PTB datasets (Marcus et al., 1994) from Hugging Face, with perplexity (PPL) as the evaluation metric. We also evaluate the pruned Llama 2-7B model using accuracy on seven Zero-Shot tasks: BoolQ (Clark et al., 2019), RTE (Wang, 2018), HellaSwag (Zellers et al., 2019), WinoGrande (Sakaguchi et al., 2021), ARC Easy and Challenge (Clark et al., 2018), and OpenbookQA (Mihaylov et al., 2018), following the experimental setup introduced by Sun et al. (2024) in their work on Wanda. This allows for a more task-specific evaluation of the pruned model's generalization capabilities. For SAM, we validate the performance using the fourth compressed file of the SA-1B dataset as well as the COCO dataset (Lin et al., 2014), employing Intersection over Union (IoU) as the primary metric for visual models.

### 4.2 UNSTRUCTURED PRUNING

**Language Models.** In Table 1 and Table 2, we report the perplexity performance of the OPT series models on Wikitext and PTB datasets after pruning to various sparsity levels using our method and other baseline approaches. Our method demonstrates superior performance over baselines even at lower sparsity levels, and as sparsity increases, the performance gap widens in favor of our approach. Additionally, as model size decreases, the advantage of our method over the baselines becomes even

more pronounced. Table 3 also presents results in comparing the performances of our proposed method against those of baselines on the Llama 2-7B model at 50% and 70% sparsity levels. Our method consistently outperforms all baseline approaches at both sparsity levels, achieving the best results across all tasks. These findings highlight the effectiveness of the inter-block error compensation pruning method introduced in this paper, particularly under high-sparsity conditions, across different model scales, and for various model architectures.

Table 1: Perplexity performances of pruned OPT models of different scales at various sparsity levels. The models are calibrated on a subset of the C4 and evaluated on full Wikitext-v2.

| Method | Model | 10% | 20% | 30% | 40% | 50% | 60% | 70% | 80% | 90% |
|---|---|---|---|---|---|---|---|---|---|---|
| Magnitude | | 28.25 | 29.65 | 34.51 | 54.60 | 193.4 | 920.0 | 3806 | 4890 | 6614 |
| SparseGPT | 125M | 27.88 | 27.98 | 28.86 | 30.56 | 37.02 | 59.55 | 220.9 | 2378 | 4666 |
| Wanda | | 27.48 | **27.76** | **28.11** | 30.67 | 38.92 | 75.17 | 328.2 | 1918 | 4609 |
| **LAMP (ours)** | | **27.37** | 27.87 | 28.66 | **29.66** | **33.90** | **45.78** | **65.20** | **270.4** | **1612** |
| Magnitude | | 14.72 | 15.62 | 24.74 | 388.0 | 1713 | 9392 | 9443 | 16344 | 28871 |
| SparseGPT | 1.3B | 14.67 | 14.75 | 15.16 | 16.49 | 17.50 | 22.08 | 51.75 | 752.4 | 6797 |
| Wanda | | 14.63 | 14.69 | 15.01 | 15.89 | 18.41 | 26.55 | 99.53 | 2258 | 16868 |
| **LAMP (ours)** | | **14.59** | **14.60** | **14.67** | **15.04** | **16.49** | **21.17** | **36.78** | **117.1** | **1099** |
| Magnitude | | 12.59 | 13.13 | 15.58 | 30.32 | 265.2 | 3604 | 7251 | 9614 | 16668 |
| SparseGPT | 2.7B | 12.32 | 12.39 | 12.66 | 12.66 | 13.46 | 16.04 | 26.92 | 138.4 | 5818 |
| Wanda | | 12.23 | 12.26 | 12.36 | 12.86 | 14.21 | 19.83 | 387.9 | 5905 | 16527 |
| **LAMP (ours)** | | **12.22** | **12.25** | **12.34** | **12.62** | **13.32** | **15.77** | **21.79** | **54.67** | **764.2** |
| Magnitude | | 10.92 | 11.27 | 12.53 | 31.89 | 968.7 | 12639 | 16975 | 25591 | 5297 |
| SparseGPT | 6.7B | 10.83 | 10.76 | 10.75 | 10.96 | 11.59 | 13.48 | 20.48 | 96.25 | 11938 |
| Wanda | | 10.73 | 10.63 | 10.64 | 10.96 | 11.98 | 15.19 | 159.2 | 3954 | 16076 |
| **LAMP (ours)** | | **10.56** | **10.58** | **10.59** | **10.85** | **11.27** | **13.03** | **17.82** | **44.69** | **1469** |

Table 2: Perplexity performancse of pruned OPT models of different scales at various sparsity levels. The models are calibrated on a subset of the C4 and evaluated on PTB.

| Method | Model | 10% | 20% | 30% | 40% | 50% | 60% | 70% | 80% | 90% |
|---|---|---|---|---|---|---|---|---|---|---|
| Magnitude | | 39.97 | 43.25 | 49.97 | 82.69 | 276.2 | 1146 | 3430 | 4124 | 5382 |
| SparseGPT | 125M | 39.43 | 40.36 | 42.71 | 45.24 | 55.78 | 88.44 | 259.6 | 2919 | 5374 |
| Wanda | | **39.05** | 39.83 | 41.43 | 44.41 | 57.59 | 109.7 | 399.7 | 2214 | 4474 |
| **LAMP (ours)** | | 39.12 | **39.13** | **40.96** | **43.47** | **49.41** | **64.10** | **88.85** | **336.3** | **3075** |
| Magnitude | | 20.66 | 21.51 | 39.89 | 823.5 | 3171 | 9499 | 8467 | 17283 | 33126 |
| SparseGPT | 1.3B | 20.34 | 20.64 | 21.50 | 23.17 | 25.47 | 33.48 | 76.91 | 531.8 | 7515 |
| Wanda | | 20.38 | 20.70 | 21.41 | 23.25 | 28.00 | 43.95 | 135.3 | 1809 | 12025 |
| **LAMP (ours)** | | **20.13** | **20.41** | **21.11** | **22.19** | **24.68** | **32.19** | **54.33** | **148.8** | **3094** |
| Magnitude | | 18.38 | 19.06 | 22.96 | 46.36 | 262.6 | 3207 | 7276 | 9575 | 12178 |
| SparseGPT | 2.7B | 18.15 | 18.50 | 18.98 | 18.98 | 20.44 | 24.67 | 42.75 | 146.0 | 6107 |
| Wanda | | **17.97** | 18.04 | 18.32 | 19.25 | 21.86 | 33.44 | 450.8 | 5462 | 21039 |
| **LAMP (ours)** | | **17.97** | **18.02** | **18.29** | **18.86** | **20.05** | **22.92** | **35.17** | **71.91** | **2357** |
| Magnitude | | 16.18 | 16.70 | 18.52 | 48.85 | 613.4 | 9379 | 9973 | 15742 | 4260 |
| SparseGPT | 6.7B | 15.77 | 15.81 | 15.92 | 16.28 | 17.43 | 20.23 | 31.86 | 105.0 | 8320 |
| Wanda | | **15.76** | 15.77 | 15.94 | 16.43 | 17.92 | 23.61 | 211.3 | 3024 | 14998 |
| **LAMP (ours)** | | **15.76** | **15.74** | **15.94** | **16.22** | **17.30** | **20.07** | **28.04** | **64.18** | **2549** |

**Vision Models.** In Tables 4 and Table 5, we report IoU performance of SAM at varying sparsity levels on SA-1B and COCO datasets with single-point interaction. Following experimental setup, SAMs are pruned with only 128 images random sampled from SA-1B as calibration data. The experiments first verify that SparseGPT indeed achieves strong IoU performance on visual models under low sparsity conditions, outperforming both Magnitude and Wanda. As Wanda is based on strong priors from language models, its underperformance compared to SparseGPT on visual models is within our expectations. Lastly, the method proposed in this paper outperforms all baselines and exhibits a similar trend observed in language model experiments: the higher the sparsity, the greater the advantage of our method over the baseline methods. This demonstrates that the proposed method is not only effective for language models but also remains effective for visual models.

Table 3: Zero-shot performance (Accuracy, %) of Llama 2-7B pruned by different method on various tasks at sparsity of 50% and 70%.

| Method | Strategy | BQ | RTE | HS | WG | ARC-e | ARC-c | OBQA | Mean |
|---|---|---|---|---|---|---|---|---|---|
| Dense | 0 | 77.74 | 63.18 | 57.10 | 68.98 | 76.26 | 43.43 | 31.40 | 59.73 |
| Magnitude | | 62.94 | **57.04** | 49.12 | 63.38 | 64.06 | 34.64 | 26.80 | 51.14 |
| SparseGPT | 50% | 76.33 | 55.96 | 52.89 | 68.98 | 72.01 | 38.23 | 28.40 | 56.11 |
| Wanda | | 76.42 | 53.43 | 52.45 | 68.67 | 72.22 | 39.33 | **31.00** | 56.22 |
| **LAMP (ours)** | | **76.50** | **57.04** | **53.03** | **69.11** | **72.39** | **40.44** | 31.00 | **56.96** |
| Magnitude | | 37.95 | 53.07 | 25.93 | 49.25 | 27.82 | 22.87 | 17.00 | 33.41 |
| SparseGPT | 70% | 64.65 | 53.79 | 33.45 | 58.64 | 43.31 | 21.67 | 17.00 | 41.79 |
| Wanda | | 48.47 | 52.71 | 27.97 | 49.64 | 30.81 | 18.60 | 12.20 | 34.34 |
| **LAMP (ours)** | | **67.71** | **55.23** | **41.45** | **60.22** | **55.56** | **27.05** | **21.40** | **46.94** |

Table 4: Instance segmentation performance of SAM models (IoU, %) at various sparsity using single-point interaction on SA-1B. The models are calibrated on a subset of 128 images sampled from SA-1B.

| Method | Model | 10% | 20% | 30% | 40% | 50% | 60% | 70% | 80% | 90% |
|---|---|---|---|---|---|---|---|---|---|---|
| Magnitude | | 74.80 | 74.57 | 73.87 | 72.26 | 69.45 | 61.51 | 28.22 | 2.65 | 1.70 |
| SparseGPT | SAM-H | 74.81 | 74.77 | 74.60 | 74.20 | 73.30 | 71.15 | 65.69 | 50.97 | 18.15 |
| Wanda | | 74.82 | 74.76 | 74.35 | 73.60 | 71.65 | 67.65 | 53.00 | 21.84 | 1.20 |
| **LAMP (ours)** | | **74.82** | **74.78** | **74.66** | **74.60** | **74.42** | **73.82** | **72.85** | **69.73** | **57.60** |
| Magnitude | | 74.81 | 74.59 | 73.93 | 72.28 | 69.18 | 62.56 | 45.52 | 9.92 | 4.52 |
| SparseGPT | SAM-L | 74.84 | 74.77 | 74.61 | 74.13 | 73.20 | 70.70 | 64.45 | 45.48 | 7.37 |
| Wanda | | **74.86** | 74.78 | 74.62 | 74.05 | 72.51 | 68.28 | 55.13 | 22.81 | 1.31 |
| **LAMP (ours)** | | 74.85 | **74.79** | **74.69** | **74.55** | **74.27** | **73.64** | **72.26** | **68.17** | **53.68** |
| Magnitude | | 72.69 | 72.56 | 71.99 | 70.36 | 67.38 | 60.87 | 46.01 | 25.53 | 1.51 |
| SparseGPT | SAM-B | **72.72** | 72.67 | 72.50 | 72.08 | 71.11 | 68.68 | 63.02 | 47.49 | 18.81 |
| Wanda | | 72.71 | 72.64 | 72.33 | 71.45 | 69.44 | 64.35 | 50.29 | 28.16 | 2.18 |
| **LAMP (ours)** | | **72.72** | **72.69** | **72.56** | **72.28** | **71.85** | **71.06** | **69.33** | **64.78** | **51.82** |

## 4.3 SEMI-STRUCTURE PRUNING AND QUANTIZATION

On certain specialized hardware, such as specific NVIDIA GPUs or other custom-designed computing architectures, the combination of pruning and quantization can further accelerate inference while maintaining an acceptable performance degradation. To explore this, we conduct tests using a joint compression strategy that combines pruning and quantization. Specifically, we focus on the combination of 2:4 and 4:8 semi-structured sparsity with 3-bit and 4-bit quantization. Since Wanda does not support quantization, we exclude it from experiments involving quantization and instead compared our approach against SparseGPT. As shown in the Table in Appendix B, our proposed method achieves the best performances, among all comparison method.

## 4.4 SPARSITY REARRANGEMENT

To validate the Sparsity Rearrangement method based on priors, as proposed in the Method section, we conducted experiments on both Inter-Block Sparsity Rearrangement and Intra-Block Sparsity Rearrangement.

**Inter-Block Sparsity Rearrangement.** In Fig. 3 and Fig. 4, we present the performance of OPT-125M and SAM-B pruned by LAMP with different $\alpha$. For clarity in visual presentation, we conducted experiments with 70% sparsity on OPT-125M and 90% sparsity on SAM-B. We observed that in the language model, $\alpha$ had the greatest impact on the performance of OPT-125M on the PTB dataset, and adjusting $\alpha$ further improved the model's performance on non-calibration datasets. On vision models, the influence of $\alpha$ on performance was similar on both datasets, with adjustments to $\alpha$ enhancing the overall performance of the visual model. Overall, appropriate tuning of $\alpha$ can further improve the performance of pruned models. Based on experiments, we conclude that $\alpha$ typically yields optimal results within the range of $[0.6, 0.75]$ for language models, and within $[0.8, 0.9]$ for vision model.

Table 5: Instance segmentation performance of SAM models (IoU, %) at various sparsity using single-point interaction on COCO.

| Method | Model | 10% | 20% | 30% | 40% | 50% | 60% | 70% | 80% | 90% |
|---|---|---|---|---|---|---|---|---|---|---|
| Magnitude | | 71.12 | 71.11 | 70.86 | 70.61 | 69.37 | 63.60 | 32.68 | 7.88 | 7.21 |
| SparseGPT | SAM-H | 71.11 | **71.18** | 70.98 | 70.93 | 70.42 | 69.52 | 66.70 | 56.82 | 27.57 |
| Wanda | | **71.17** | 71.01 | 70.98 | 70.73 | 69.63 | 66.05 | 53.45 | 22.83 | 5.51 |
| **LAMP (ours)** | | 71.14 | **71.18** | **71.12** | **70.98** | **70.81** | **70.24** | **69.43** | **66.69** | **58.41** |
| Magnitude | | **70.56** | 70.51 | 70.27 | 69.53 | 68.08 | 62.66 | 49.12 | 20.66 | 18.12 |
| SparseGPT | SAM-L | 70.55 | 70.53 | 70.43 | 70.04 | 69.61 | 68.21 | 64.42 | 48.35 | 11.30 |
| Wanda | | **70.56** | **70.58** | 70.45 | 69.91 | 68.65 | 65.56 | 54.67 | 19.89 | 6.38 |
| **LAMP (ours)** | | 70.55 | **70.58** | **70.58** | **70.37** | **69.91** | **69.51** | **68.14** | **65.42** | **56.92** |
| Magnitude | | 66.84 | **67.06** | **67.06** | 66.67 | 65.29 | 62.28 | 53.61 | 40.86 | 7.48 |
| SparseGPT | SAM-B | 67.09 | 67.02 | 66.98 | 66.76 | 66.17 | 65.00 | 62.03 | 52.46 | 28.51 |
| Wanda | | 66.94 | 66.97 | 67.04 | 66.47 | 65.38 | 62.31 | 53.58 | 39.32 | 7.89 |
| **LAMP (ours)** | | **67.10** | **67.06** | 67.04 | **66.83** | **66.45** | **66.07** | **64.51** | **62.40** | **54.50** |

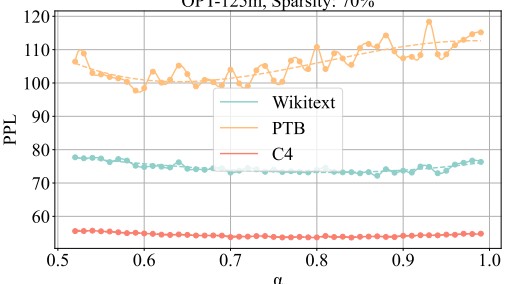 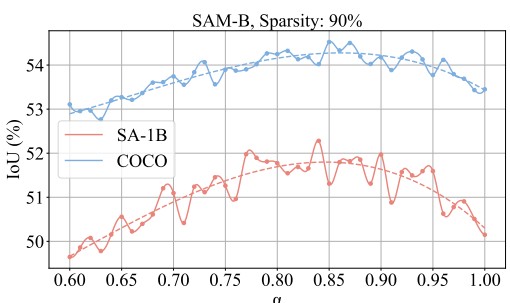

Figure 3: Perplexity performance of pruned OPT-125M with different $\alpha$.

Figure 4: Instance segmentation performance (IoU, %) of SAM-B with different $\alpha$.

**Intra-Block Sparsity Rearrangement.** In Fig. 5, we illustrate the impact of different $\beta$ on the language model, using the 70% sparsity model of OPT-125M as an example. We observed that in the language model, adjusting $\beta$ can further reduce the model's perplexity. Unlike $\alpha$, $\beta$ not only improves the model's performance on non-calibration datasets but also enhances its performance on calibration datasets. Overall, appropriately adjusting $\beta$ can further improve the performance of the pruned model. The optimal value for $\beta$ typically falls within the range of $[0.6 \times (1-s), 0.8 \times (1-s)]$, where $s$ denotes sparsity. In Fig. 6, we exemplify the effects of varying $\beta$ on the visual model, specifically using the 90% sparsity model of SAM-B. Our findings indicate that the influence of $\beta$ on the visual model is minimal; both on calibration and non-calibration datasets, adjustments to $\beta$ yield only marginal improvements in the model's performance.

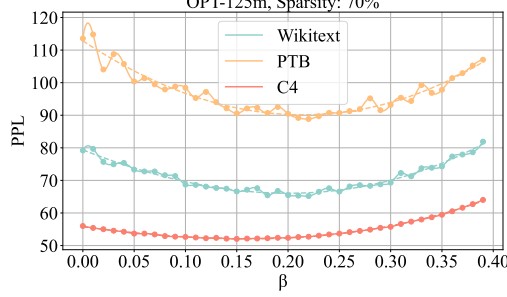 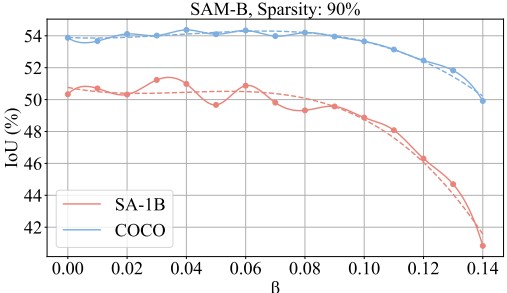

Figure 5: Perplexity performance of OPT-125M with different $\beta$.

Figure 6: Instance segmentation performance (IoU, %) of SAM-B with different $\beta$.

### 4.5 CALIBRATION SET SIZE

Although our experiments follow the settings of SparseGPT and Wanda, using 128 randomly selected samples for pruning both language and vision models, Fig. 7 and Fig. 8 show the impact of calibration set size on our method's performance. Fig. 1 presents results on the OPT-125M language model at 70% sparsity, while Fig. 2 shows results for the SAM-B vision model at 90% sparsity. In both cases, increasing the sample size beyond 128 offers minimal performance gains while significantly increasing GPU memory usage. Thus, like SparseGPT and Wanda, 128 samples remain the optimal choice.

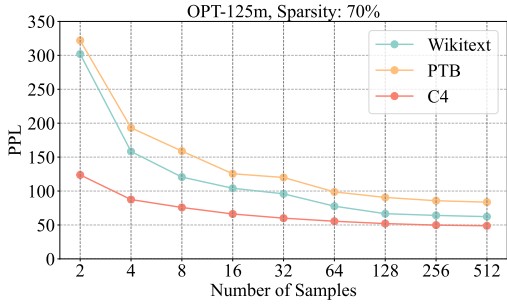

Figure 7: Perplexity performance of OPT-125M with different calibration set size.

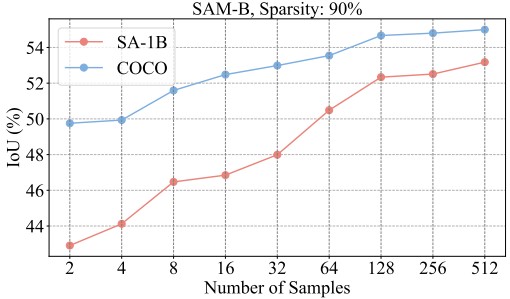

Figure 8: Instance segmentation performance (IoU, %) of SAM-B with different calibration set size.

### 4.6 GPU MEMORY CONSUMPTION

In Table 6, we report the GPU memory consumption of SparseGPT, Wanda, and our method when pruning the OPT-6.7B. For large model pruning algorithms, GPU memory consumption directly determines whether the method can be executed on devices with lower memory capacity and with lower cost. Our analysis indicates that the additional GPU memory usage in SparseGPT and Wanda, compared to the proposed method, primarily comes from stor-

Table 6: GPU Memory Consumption Comparison

| Method | GPU Memory Consumption |
|---|---|
| SparseGPT | 8.3 GB |
| Wanda | 21 GB |
| **LAMP (ours)** | **7.8 GB** |

ing intermediate results. However, the results show that, because of chunked intermediate storage, our method, even though it involves partial model training, has lower GPU memory consumption than both SparseGPT and Wanda. Note that we only use mixed-precision fine-tuning, without employing low-rank adaptation methods. If combined with efficient fine-tuning techniques such as LoRA, the GPU memory consumption could be further reduced. This advantage enables the pruning of larger models on consumer-grade GPUs.

## 5 CONCLUSION

In this paper, we propose an efficient pruning method, LAMP, which is effective for both vision and language models while consuming limited GPU memory. The LAMP method fine-tunes each block individually by compensating for the inter-block errors introduced by pruning, then rearranges the sparsity distribution based on the prior knowledge of the Transformer structure and pruning method. Additionally, system memory is utilized to offload the GPU memory burden of storing intermediate results. The proposed method significantly improves the performance of models pruned at high sparsity levels without increasing GPU memory consumption. We validate the LAMP's effectiveness across vision and language models of varying scales, achieving state-of-the-art performance in both domains.

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

## A  THEORETICAL FOUNDATIONS

Large models are typically hierarchical and can be effectively viewed as a complex nested function:

$$F(x_i; \theta) = f_n \left( \dots f_j \left( \dots f_1 \left( x; \theta_1 \right) \dots; \theta_j \right) \dots; \theta_n \right), \tag{5}$$

where $x$ is the input sample, $f_j$ represents the mapping function of Block $B^j$ and $\theta$ denotes the function parameters. Pruning a block effectively introduces a perturbation $\Delta\theta$ to that block's parameters. Taking $B^j$ as an example, when pruning is applied such that $\theta_j = \theta_j + \Delta\theta$, the error introduced by this perturbation propagates from the output of $B^j$ through to the final Block, leading to a deviation in the global model output:

$$\Delta\mathcal{Y}_j \approx \frac{\partial F}{\partial f_j} \cdot \frac{\partial f_j}{\partial \theta_j} \cdot \Delta\theta_j = \left( \prod_{i=j+1}^{n} \frac{\partial f_i}{\partial z^{i-1}} \right) \cdot \frac{\partial z^j}{\partial \theta_j} \cdot \Delta\theta_j, \tag{6}$$

where $\partial z^j$ represents the output of $B^j$, in the model, and $\Delta\mathcal{Y}_j$ denotes the change in the model output caused by pruning $B^j$. From this equation, we observe that the closer the pruned block is to the model's output (*i.e.*, the further back in the model), the more directly the pruning affects the output. Conversely, the further the pruned block is from the output, the more difficult it becomes to control the resulting error. However, we can select a subsequent Block $B^k$ after $B^j$ to compensate for the changes introduced by pruning. If we adjust the parameters of $B^k$, its effect on the model output is similarly given by:

$$\Delta\mathcal{Y}_k = \left( \prod_{i=k+1}^{n} \frac{\partial f_k}{\partial z_{k-1}} \right) \cdot \frac{\partial z^k}{\partial \theta_k} \cdot \Delta\theta_k. \tag{7}$$

By setting $\Delta\mathcal{Y}_j + \Delta\mathcal{Y}_k = 0$, we can derive:

$$\Delta\theta_k = - \frac{\left( \prod_{i=j+1}^{n} \frac{\partial f_i}{\partial z^{i-1}} \right) \cdot \frac{\partial f_j}{\partial \theta_j} \cdot \Delta\theta_j}{\left( \prod_{i=k+1}^{n} \frac{\partial f_i}{\partial z^{i-1}} \right) \cdot \frac{\partial f_k}{\partial \theta_k}}. \tag{8}$$

From Eq. (8), we can observe that the closer $B^k$ is to $B^j$, the more direct the relationship between their parameters. Specifically, if $k = j + 1$, then $\prod_{i=j+1}^{k} \frac{\partial f_i}{\partial z^{i-1}} = \frac{\partial f_{j+1}}{\partial z^j}$, making the compensation more direct and easier to control.

However, the above compensation method requires access to the model's output, which can be both memory-intensive and time-consuming. Therefore, Progressive Block Pruning adopts a partial compensation approach. During error compensation, we directly use the MSE loss of $z^k$, the output of $B^k$, as the compensation loss. This can be expressed as:

$$\mathcal{L}_k(\theta_k) = \frac{1}{2m} \sum_{i=1}^{m} \| z'^k(\theta'_k) - z^k(\theta_k) \|^2. \tag{9}$$

The global optimization objective for the model is:

$$\mathcal{L}_{global} = \frac{1}{2m} \sum_{i=1}^{m} \| F'(x_i) - F(x_i) \|^2. \tag{10}$$

The optimization directions of Eq. (9) and Eq. (10) are aligned because, when we take the derivatives of these two equations, we obtain:

$$\frac{\partial \mathcal{L}_k(\theta_k)}{\partial \Delta\theta_k} = \frac{1}{m} \sum_{i=1}^{m} \left( \Delta z'^k(x_i) \cdot \frac{\partial z^k}{\partial \theta_k} \right), \tag{11}$$

$$\frac{\partial \mathcal{L}_{global}}{\partial \Delta\theta_k} = \frac{1}{m} \sum_{i=1}^{m} \left( \frac{\partial F}{\partial z^k} \cdot \Delta z'^k(x_i) \cdot \frac{\partial z^k}{\partial \theta_k} \cdot \frac{\partial F}{\partial z^k} \right). \tag{12}$$

The signs of these derivatives are the same, indicating that the optimization directions are consistent. Therefore, we can use $\mathcal{L}_k$ to implicitly optimize $\mathcal{L}_{global}$.

Based on the above theoretical analysis, we can draw the following three conclusions:

- The error introduced by pruning can be compensated for by adjusting subsequent blocks in the model.

- The closer the compensating block is to the pruned block, the more direct the relationship between their parameters.

- Local MSE can be used to implicitly optimize the global MSE.

## B  SEMI-STRUCTURE PRUNING AND QUANTIZATION

In Section 4.3 of the main text, we discussed the experiments conducted on both language models and vision models using a joint pruning-quantization compression strategy. Specifically, we applied four distinct compression strategies to the OPT, Llama 2-7B, and SAM models: 2:4 and 4:8 semi-structured pruning, as well as a combination of semi-structured pruning with 3-bit and 4-bit quantization. In Table 7, we present the performance of different methods on the OPT series models under various model compression strategies. Our method outperforms the baselines in both semi-structured pruning and the combined pruning-quantization compression strategy. Similarly, in Tables 8 and 9, we report the results of identical experiments conducted on the Llama 2-7B and SAM series vision models, respectively, where we observed consistent improvements.

Table 7: Perplexity performance of pruned OPT models at various compression strategies.

| Method | Strategy | OPT-125M | | OPT-1.3B | | OPT-2.7B | | OPT-6.7B | |
|---|---|---|---|---|---|---|---|---|---|
| | | Wikitext | PTB | Wikitext | PTB | Wikitext | PTB | Wikitext | PTB |
| SparseGPT | 2:4 | 59.35 | 93.04 | 23.87 | 38.09 | 17.11 | 26.98 | 14.15 | 21.54 |
| Wanda | 2:4 | 80.01 | 111.97 | 28.20 | 43.37 | 21.18 | 34.55 | 15.90 | 25.09 |
| **LAMP (ours)** | 2:4 | **46.90** | **67.56** | **22.03** | **33.95** | **16.39** | **25.31** | **14.04** | **21.19** |
| SparseGPT | 4:8 | 43.90 | 71.99 | 20.21 | 31.36 | 15.00 | 23.05 | 12.49 | 18.86 |
| Wanda | 4:8 | 53.18 | 79.02 | 22.17 | 34.56 | 16.79 | 26.13 | 13.55 | 20.19 |
| **LAMP (ours)** | 4:8 | **39.91** | **59.34** | **19.15** | **30.41** | **14.88** | **22.88** | **12.38** | **18.53** |
| SparseGPT | 2:4+int3 | 122.33 | 196.44 | 40.58 | 64.53 | 24.70 | 41.06 | 20.07 | 31.91 |
| **LAMP (ours)** | 2:4+int3 | **74.07** | **114.93** | **27.88** | **46.14** | **19.57** | **30.28** | **18.38** | **26.82** |
| SparseGPT | 2:4+int4 | 70.88 | 107.39 | 25.96 | 41.69 | 18.56 | 30.68 | 15.24 | 23.01 |
| **LAMP (ours)** | 2:4+int4 | **51.24** | **75.39** | **23.49** | **34.42** | **17.05** | **27.04** | **15.10** | **22.54** |
| SparseGPT | 4:8+int3 | 84.02 | 120.16 | 32.41 | 49.27 | 20.24 | 32.99 | 17.08 | 26.36 |
| **LAMP (ours)** | 4:8+int3 | **59.67** | **86.01** | **24.73** | **40.19** | **17.89** | **26.80** | **16.16** | **24.30** |
| SparseGPT | 4:8+int4 | 50.58 | 80.97 | 23.32 | 34.38 | 15.74 | 24.76 | 13.44 | 19.72 |
| **LAMP (ours)** | 4:8+int4 | **42.03** | **60.60** | **20.86** | **31.58** | **15.33** | **23.80** | **13.24** | **19.49** |

Table 8: Zero-shot Performance (Accuracy, %) of Llama 2-7B Model under various compression strategies.

| Method | Strategy | BQ | RTE | HS | WG | ARC-e | ARC-c | OBQA | Mean |
|---|---|---|---|---|---|---|---|---|---|
| SparseGPT | 2:4 | 68.07 | 58.84 | 43.38 | **65.59** | 64.18 | 31.91 | 24.80 | 50.97 |
| Wanda | 2:4 | 68.13 | 53.43 | 41.42 | 62.43 | 63.13 | 30.63 | 23.80 | 49.00 |
| **LAMP (ours)** | 2:4 | **69.72** | **58.84** | **46.75** | 64.48 | **66.04** | **33.70** | **26.20** | **52.25** |
| SparseGPT | 4:8 | 71.07 | 54.87 | 48.35 | **67.80** | 68.69 | 34.47 | 27.20 | 53.21 |
| Wanda | 4:8 | 73.00 | 53.79 | 47.03 | 66.93 | 67.47 | 34.22 | 27.00 | 52.78 |
| **LAMP (ours)** | 4:8 | **73.12** | **58.12** | **50.13** | 64.88 | **68.90** | **36.01** | **28.00** | **54.16** |
| SparseGPT | 2:4+int3 | 64.65 | **55.23** | 38.58 | 59.04 | 55.47 | 25.34 | 19.40 | 45.39 |
| **LAMP (ours)** | 2:4+int3 | **66.82** | 54.15 | **43.36** | **60.77** | **59.55** | **28.67** | **21.00** | **47.76** |
| SparseGPT | 2:4+int4 | 66.73 | 54.15 | 42.34 | **65.19** | 61.74 | 28.92 | **23.80** | 48.98 |
| **LAMP (ours)** | 2:4+int4 | **70.46** | **56.68** | **46.22** | 62.90 | **64.81** | **30.63** | 22.80 | **50.64** |
| SparseGPT | 4:8+int3 | 68.59 | 53.43 | 42.12 | 62.35 | 60.10 | 29.10 | 21.60 | 48.18 |
| **LAMP (ours)** | 4:8+int3 | 68.38 | **54.87** | **43.54** | **62.85** | **63.38** | **32.34** | **22.40** | **49.68** |
| SparseGPT | 4:8+int4 | 69.27 | 54.15 | 46.58 | **66.06** | 67.55 | **34.39** | 27.20 | 52.17 |
| **LAMP (ours)** | 4:8+int4 | **72.29** | **61.01** | **48.86** | 62.75 | **66.88** | 33.87 | **28.00** | **53.38** |

Table 9: Instance segmentation performance of SAM models (IoU, %) at various compression strategies using single-point interaction on SA-1B and COCO.

| | | SAM-B | | SAM-L | | SAM-H | |
|---|---|---|---|---|---|---|---|
| Method | Strategy | COCO | SA-1B | COCO | SA-1B | COCO | SA-1B |
| SparseGPT | 2:4 | 62.80 | 67.59 | 65.01 | 69.07 | 67.02 | 69.62 |
| Wanda | 2:4 | 60.59 | 62.06 | 62.37 | 65.84 | 64.68 | 66.36 |
| **LaMP(ours)** | 2:4 | **64.95** | **71.30** | **67.83** | **73.56** | **69.09** | **73.64** |
| SparseGPT | 4:8 | 63.93 | 69.69 | 66.56 | 71.27 | 67.98 | 71.41 |
| Wanda | 4:8 | 62.62 | 66.05 | 65.02 | 69.75 | 66.80 | 69.55 |
| **LaMP(ours)** | 4:8 | **65.37** | **71.72** | **68.17** | **73.86** | **69.44** | **74.03** |
| SparseGPT | 2:4+int3 | 62.09 | 64.77 | 63.45 | 65.67 | 66.39 | 67.85 |
| **LaMP(ours)** | 2:4+int3 | **64.47** | **69.71** | **67.13** | **72.60** | **68.75** | **73.26** |
| SparseGPT | 2:4+int4 | 62.66 | 66.76 | 64.78 | 68.69 | 66.81 | 69.44 |
| **LaMP(ours)** | 2:4+int4 | **64.89** | **70.64** | **67.64** | **73.09** | **69.02** | **73.72** |
| SparseGPT | 4:8+int3 | 63.02 | 67.27 | 64.74 | 68.02 | 67.25 | 69.69 |
| **LaMP(ours)** | 4:8+int3 | **64.84** | **69.86** | **67.68** | **73.28** | **69.10** | **73.35** |
| SparseGPT | 4:8+int4 | 63.77 | 68.98 | 66.04 | 70.31 | 67.87 | 71.05 |
| **LaMP(ours)** | 4:8+int4 | **65.24** | **70.98** | **68.02** | **73.82** | **69.34** | **73.78** |

