# OpenReview forum: "LAMP: Large Model Pruning with Inter-Block Error Compensation"
_ICLR.cc/2025/Conference — ICLR 2025 Conference Withdrawn Submission_

### Official Review · Reviewer_7VsB · 2024-10-28

**Soundness:** 3
**Presentation:** 3
**Contribution:** 2
**Rating:** 5
**Confidence:** 4

**Summary:**

The paper presents LAMP, a pruning technique designed for large models in both vision and language tasks. It focuses on addressing memory and resource constraints associated with pruning, particularly at high sparsity levels. Traditional pruning methods often struggle with performance degradation at these levels or require extensive fine-tuning. In contrast, LAMP introduces a sliding window-based, progressive block pruning method that compensates for errors between blocks, minimizing memory use and preserving model performance even at high sparsity.

**Strengths:**

1. The core innovation of LAMP is to compensate for the pruning error between blocks, by fine-tuning the next block to compensate for the pruning error of the current block. This allows AMP to effectively maintain model performance even at high sparsity levels, making it suitable for resource-constrained environments.

2. This technique is applicable to both visual and language models, and the authors have demonstrated its robustness across domains and tasks, with improvements in language models (such as OPT and Llama 2) and visual models (such as SAM). Compared to previous pruning methods for large models, this work is not limited to exploring LLMs, but also extends to visual segmentation models, showcasing its broader applicability.

3. The authors provide extensive experiments to validate the effectiveness of LAMP on multiple metrics, and compares it with previous methods such as SparseGPT and Wanda on metrics including perplexity, accuracy, and IoU (Intersection over Union).

**Weaknesses:**

1. The construction of block error compensation is not detailed enough. The core of LAMP is to reduce the impact of pruning through block error compensation, but the authors only mention in **Section 4.6** that they used **mixed-precision fine-tuning**, without any discussion or explanation of the tuning gradient changes of each block and hyperparameter selection (Algorithm 1, line 8), which is confusing. Moreover, using block error compensation for large model compression is not a new technique. CBQ[1] has used a similar framework to compensate for the compression error of LLMs. The sliding block error compensation technique has been attempted in previous work on LLM compression, but this paper does not discuss or cite this work.

2. The authors mainly compared LAMP with Wanda and SparseGPT, two LLM pruning works, but SparseGPT did not use any training or fine-tuning to correct the pre-trained model. However, LAMP requires progressive tuning to compensate for errors, although the GPU consumption of block-level tuning is not high for models with 7B or fewer parameters, the authors did not discuss the time-consuming problem and compression efficiency brought by this technique, which is unfair. At the same time, the scalability of this method to larger models is questionable. Can it compress a 70B model with the same efficiency? The authors did not provide any discussion or experiments to demonstrate the scalability of LAMP, which is an important aspect of a pruning method. This lack of discussion and experimentation raises concerns about the practicality and applicability of LAMP to larger models.

3. Similar to Weakness 2, Wanda discussed both the completely untrained pruning method and the method with full fine-tuning and Lora fine-tuning to compensate for pruning. LAMP uses block-level tuning to compare with Wanda, but in this manuscript, it is not explicitly stated whether the comparison is with the untuned Wanda or the Wanda with tuning compensation. This point is also confusing. It is unclear whether the authors are comparing LAMP with the baseline Wanda method or with the Wanda method that has been fine-tuned to compensate for pruning errors. This lack of clarity makes it difficult to understand the true performance of LAMP and its comparison with Wanda. The authors should provide a clear explanation of the comparison setup to avoid confusion.

4. The variable $z_j^{2'}$ in line 8 of **Algorithm 1** is not defined anywhere in the algorithm, and the calculation in this line contradicts the explanation in **Figure 2** and **Appendix A**. This inconsistency raises concerns about the correctness and clarity of the algorithm.

5. The two hyperparameters, \alpha and \beta, will simultaneously affect the output performance of the model. Discussing them separately has neglected their combined impact (Figure 3/4, Figure5/6).


[1]CBQ: Cross-Block Quantization for Large Language Models

**Questions:**

1. The Equation (1) and (2) and the explanation suggest that the last block is processed by equally increasing the sparsity of the other blocks, but the sensitivity and impact of different blocks on the output are not the same in LLM and SAM. Generally, the earlier layers and the last few blocks are more sensitive to the output. Is the equal block sparsity strategy in Equation (2) the optimal strategy?

---

### Official Review · Reviewer_frpX · 2024-10-31

**Soundness:** 3
**Presentation:** 2
**Contribution:** 2
**Rating:** 3
**Confidence:** 4

**Summary:**

This paper proposes a pruning method for large language and vison models. Progressive block pruning is proposed to reduce the pruning error by enhancing error compensation. Inter-block and Intra-block sparsity rearrangements apply different sparsity level to different layers and weights. The experimental results show better pruning performance on OPT, Llama and SAM models.

**Strengths:**

1.	The proposed method works better than SparseGPT and Wanda in high sparsity level.
2.	Good related works explanations.

**Weaknesses:**

1.	The novelty of this paper is low, and all methods are very incremental. 1) The layer error propagation problem has been considered in previous works. 2) Sparsity rearrangements have been proposed in the work [1]. 3) The GPU memory offloading method is also a common method.
2.	The evaluation is not comprehensive. 1) Except SparsGPT and Wanda, Pruner-Zero [2] is SOTA pruning method, and this paper didn’t compare with it. 2) More vision models should be evaluated to show the pruning performance on vision tasks. 3) The ablation studies at section 4.4, 4.5 should evaluate more large and general models rather than only on opt-125m. 4) The incremental performance for block pruning, sparsity rearrangements should be evaluated.
3.	The writing of this paper should be improved. 1) The description of algorithm 1 is hard to understand. 2) There should be more data analysis in the experiments.

**Questions:**

1.	The paper aims at pruning methods on vision and language models. But what’s the key difference for pruning between vision and language models? It seems that the SparseGPT and Wanda which design for language models show similar performance on vision models.
2.	Do the proposed pruning methods work on multi-modal models?
3.	There are two layers in one block in this paper. What about setting more layers in one pruning block?
4.	This paper shows better performance on high sparsity pruning tasks. But the model performance is still unacceptable. What’s the meaning to set the very high sparsity here?

---

### Official Review · Reviewer_9gGh · 2024-11-02

**Soundness:** 2
**Presentation:** 2
**Contribution:** 2
**Rating:** 5
**Confidence:** 4

**Summary:**

The paper proposes a pruning method aimed at improving model performance at high sparsity levels. The authors introduce a novel inter-block error compensation strategy to address performance degradation in pruned models and claim improvements over existing methods like SparseGPT and Wanda.

**Strengths:**

1.  The sliding window approach and inter-block compensation mechanism are described in detail. The authors present a thorough explanation of their technique, supplemented by theoretical analysis.

2. The experiments cover a range of models and tasks, demonstrating the method’s effectiveness in both language and vision domains. The performance gains are highlighted and show some promise.

**Weaknesses:**

1.  Although the paper claims efficient memory usage, it does not sufficiently address the computational overhead introduced by the error compensation mechanism. The trade-off between performance gains and increased complexity is not convincingly analyzed, making it hard to assess the method’s practicality.

2. The novelty of the proposed method is kind of incremental. And the "progressive" strategy has been widely used in model compression.

3. Some of the improvement of experimental results over baselines are marginal when the pruning ratio under 50%.

4. I have some reproducibility concerns, some experimental details are missing, and there is no code available.

**Questions:**

Please refer to the weakness.

---

### Official Review · Reviewer_raXG · 2024-11-05

**Soundness:** 1
**Presentation:** 1
**Contribution:** 2
**Rating:** 1
**Confidence:** 4

**Summary:**

The paper presents LAMP, a pruning method for large models that introduces inter-block error compensation to improve performance at high sparsity levels, applied to both vision and language models. However, the paper lacks rigor in both presentation and experimental validation, which undermines the impact of the proposed contributions.

**Strengths:**

Addressing the challenge of pruning large models for efficiency is a significant research area with high applicability. Given the increased demand for deploying large-scale models on limited resources, any advancements in pruning techniques could have a substantial impact. The concept of inter-block error compensation may have potential for broader application if rigorously validated.

**Weaknesses:**

- Lack of Clarity in Explanation
- Limited Practical Relevance of Results: The primary improvements achieved are at high perplexity levels, where model performance is already suboptimal. This raises questions about the practical utility of the results, as improvements in more practical, moderate-sparsity settings would likely be of greater interest to practitioners.
- Narrow Experimental Scope
- Insufficient Validation of Key Assumptions: The inter-block error compensation concept is interesting but inadequately validated. An ablation study on the specific effect of inter-block compensation on error propagation would be valuable in assessing its true impact. This would be especially relevant for understanding the broader applicability of the approach across different architectures and sparsity levels.
- Underdeveloped Analysis for Vision Models: While LAMP is tested on vision models, the results and analysis are limited, and the applicability to vision models is not fully explored.

**Questions:**

How does the performance of LAMP compare at moderate sparsity and perplexity levels, which are more common in real-world applications? It would be useful to see results that reflect typical deployment conditions rather than focusing solely on high-sparsity scenarios.

---

### Note · Authors · 2024-11-15

I have read and agree with the venue's withdrawal policy on behalf of myself and my co-authors.